# MoE Meets Reparameterization: Reparameterizable Mixture-of-Experts Model Enhances Dermatology Diagnosis via Dense-to-Experts Distillation

## Abstract

Reliable automated dermatological assessment hinges on algorithms capable of operating consistently amid the substantial heterogeneity inherent in clinical imaging data. A robust foundation model with strong representational flexibility to capture such complex variability is crucial. Here, we introduce SkinMoE, the first Mixture-of-Experts (MoE)–based foundation model specifically designed for dermatology image analysis. At the core of SkinMoE is Dense-to-Experts Distillation (DED), a novel knowledge distillation strategy that transfers rich representations from a pretrained dense vision transformer, PanDerm, to a set of expert networks. A key innovation is the Mergeable-MoE Block, which enables joint expert training and reparameterization into a single 1×1 convolution at inference time, preserving the computational efficiency of standard feed-forward networks. Unlike prior MoE approaches that use sparse top-k routing, SkinMoE employs a soft weighting mechanism, allowing all experts to contribute to predictions. This design enhances model expressiveness while introducing only a negligible increase in inference computation, regardless of the number of experts. On the DermNet dataset, SkinMoE achieves up to a 2.5% improvement in Weighted F1 over PanDerm. Ablation studies confirm the contributions of each component. Code and pretrained models will be released.

## 1 Introduction

Dermatology image analysis (Esteva et al., 2017; Azizi et al., 2021; Barata et al., 2023) faces a fundamental obstacle that is rarely present at the same scale in other medical modalities. Images originate from heterogeneous cameras and smartphones, are taken under uncontrolled illumination, include diverse skin tones, and capture lesions that vary widely in size, morphology, and anatomical context. These factors jointly induce pronounced distribution shifts across datasets and clinical sites (Fogelberg et al., 2023; Jeong et al., 2023; Wu et al., 2022). Empirical evidence demonstrates that models trained on a single cohort suffer marked performance degradation when evaluated on external collections, even when the label space is identical (Chamarthi et al., 2024).

Leveraging large scale pre-training on millions of heterogeneous dermatology images, contemporary vision backbones substantially mitigate the problem described above and now reach performance that rivals experienced clinicians across a wide spectrum of tasks (Shen et al., 2024b). The recently released PanDerm (Yan et al., 2025), trained on more than two million images, exemplifies this progress and delivers strong finetuned results on numerous public benchmarks. However PanDerm remains a dense monolithic network that optimises a single set of shared parameters for all inputs. Consequently it lacks the structural flexibility required to adapt to the wide variety of skin tones imaging devices and lesion morphologies encountered in routine clinical settings. This limitation leaves considerable room for improving robustness and enhancing domain specific representation capacity.

Mixture-of-Experts (MoE) architectures partition model capacity into specialised expert subnetworks and enable conditional computation through data-dependent routing. The original sparsely

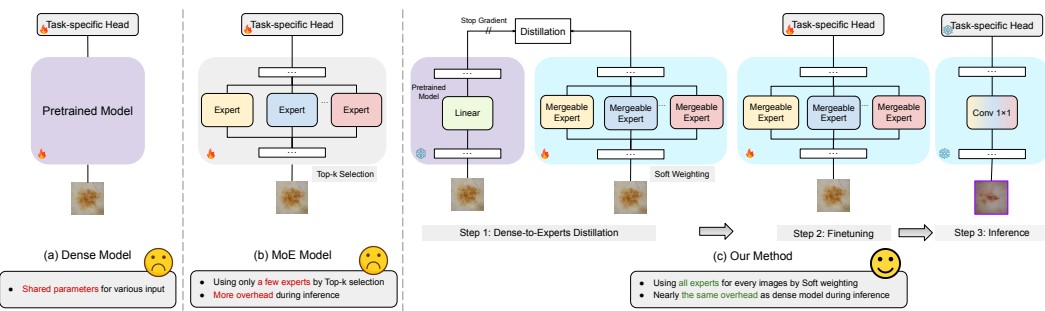

Figure 1: Comparison of our method with previous methods. (a) A dense backbone that deploys a single parameter set for all inputs. (b) A conventional Mixture-of-Experts that activates only a few of experts through hard top $k$ routing. (c) Our proposed pipeline, which distils knowledge into mergeable experts, applies soft-weighted fine-tuning and reparameterises each block into a single $1 \times 1$ convolution during inference, thereby combining full expert utilisation with dense-model efficiency.

gated design in language modelling (Shazeer et al., 2017) has evolved into trillion-parameter systems such as Switch Transformer (Fedus et al., 2022), GShard (Lepikhin et al., 2020) and GLaM (Du et al., 2022). More studies further mitigate load imbalance by refining gating strategies and usage regularisation (Zoph et al., 2022; Xue et al., 2024; Jin et al., 2025; Zhou et al., 2022; Liu et al., 2022). When transplanted to vision, MoE principles reinforce Transformer backbones including V-MoE (Riquelme et al., 2021), M³ViT (Fan et al., 2022), ViMoE (Han et al., 2024) and Mobile V-MoE (Daxberger et al., 2023), delivering dense-level accuracy with reduced average computation. Despite these advances most visual MoE designs still rely on sparse top-k routing that selects only a small subset of experts, thereby restricting the utilisation of the model's full representational capacity. Moreover prior studies concentrate on generic benchmarks, and no same-scale MoE backbone has yet been investigated for dermatology, where heterogeneous skin tones imaging devices and lesion morphologies create pronounced sub-domain shifts.

To solve these problems, we propose SkinMoE, the first dermatology specific Mixture of Experts backbone that preserves the parameter budget of its dense predecessor while markedly enhancing its capacity to model heterogeneous inputs. Our pipeline begins with Dense-to-Experts Distillation (DED), a teacher student distillation stage that transfers PanDerm's rich representations to a pool of tentative expert weights (Hinton et al., 2015). Guided by this task and building upon the reparameterisation (Ding et al., 2019; Guo et al., 2020; Ding et al., 2021a; Vasu et al., 2023b) paradigm, we design a Mergeable-MoE Block which contains several experts that can be initialised with the corresponding layer parameters in the teacher PanDerm. The experts are then jointly optimised under a learnable soft weighting scheme that replaces top $k$ routing and allows every expert to influence each prediction. During inference, the experts and their router can be reparameterised (Ding et al., 2021c;b; Vasu et al., 2023a; Li et al., 2022) into a single $1 \times 1$ convolution in order to maintain inference efficiency. The combined effect of informed distillation and fully activated expert diversity enables SkinMoE to deliver superior accuracy in cross dataset evaluations while retaining the original model size and computational footprint. Our contributions are fourfold:

- We present SkinMoE, the first Mixture of Experts backbone specifically designed for dermatology image analysis.

- We introduce Dense-to-Experts Distillation that transfers the knowledge of a large pretrained vision transformer PanDerm into a multi expert system, representing the first attempt to build an MoE model distilled from a domain specific foundation model.

- We design the Mergeable-MoE Block, which employs a differentiable soft weighting router to engage every expert during training and can be reparameterised into a single $1 \times 1$ convolution for efficient inference.

- Extensive evaluations across multiple datasets and downstream tasks confirm the effectiveness of our approach, showing that SkinMoE yields substantial performance gains while maintaining nearly identical FLOPs to PanDerm.

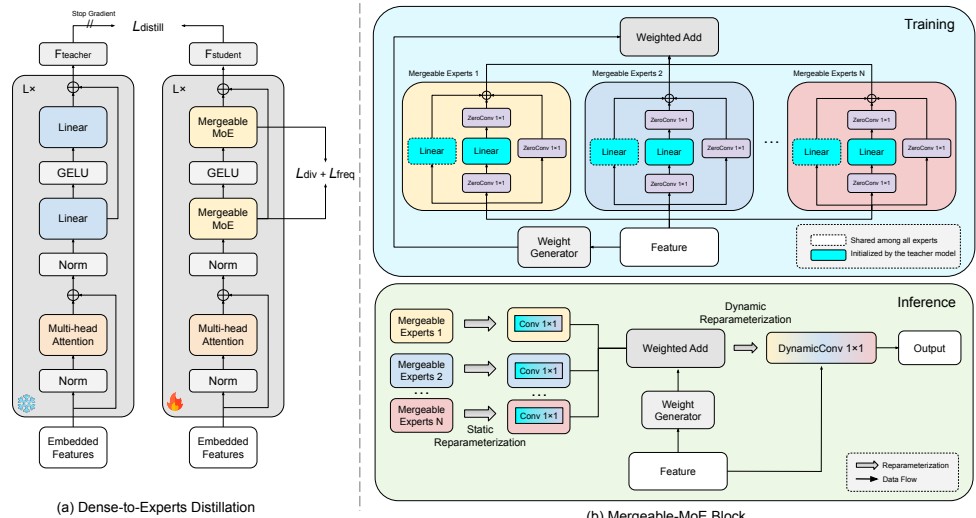

Figure 2: Overview of SkinMoE. (a) Dense-to-Experts Distillation aligns the frozen PanDerm teacher with a student model that replaces each linear transformation inside the feed-forward networks with a Mergeable-MoE Block. (b) Each block hosts multiple experts with a soft-weighting router during training and all branches are later fused into a single $1 \times 1$ kernel for efficient inference.

## 2 METHOD

### 2.1 OVERVIEW

As illustrated in Figure 2, the proposed SkinMoE framework augments dermatology image modelling by uniting Dense-to-Experts Distillation (DED) with a reparameterisable Mergeable-MoE Block. This design pursues two complementary objectives. First, it enlarges representational diversity through multiple experts that can specialise to distinct visual sub domains present in heterogeneous clinical data. Second, it keeps the computational footprint during the inference stage comparable to that of the original dense backbone.

During training the dense teacher network PanDerm operates in a teacher–student configuration. The outputs of the teacher model are transferred to the student experts through DED. Each Mergeable-MoE Block comprises several experts that are coordinated by a learnable soft weighting router. The router assigns continuous weights to all experts for every input sample so that every expert contributes to the forward pass. This full participation contrasts with sparse top $k$ routing and encourages the experts to learn complementary features while avoiding capacity under utilisation.

During the inference stage, each Mergeable-MoE Block undergoes a two step reparameterisation. Firstly, within each expert all branches which contain linear layer and zero initialised convolutional are algebraically merged into a single static $1 \times 1$ convolution kernel. Secondly, the weighted sum of these static kernels is then computed according to the router output yielding one dynamic $1 \times 1$ kernel per block. Consequently, the forward pass of SkinMoE involves nearly the same computational cost as our baseline model PanDerm, apart from the minor overhead introduced by the lightweight router. In this manner SkinMoE attains improved domain adaptive capacity while maintaining near original inference efficiency.

### 2.2 DENSE-TO-EXPERTS DISTILLATION

Dense-to-Experts Distillation is designed to migrate the knowledge encoded by a large pretrained vision transformer into a Mixture-of-Experts student. The dense PanDerm backbone serves as the frozen teacher, while the student SkinMoE replaces every linear layer in each dense feed-forward network with a Mergeable-MoE Block that comprises $E$ experts. Model initialisation follows two rules. First, for layers that are not part of an MoE block, their weights are copied directly from teacher to student so that identical computations are obtained at start-up. Second, inside every

Mergeable-MoE Block (as introduced in the next subsection), the linear components of each expert are initialised with the corresponding teacher layers $\mathcal{F}_1$ and $\mathcal{F}_2$, while all convolutional augmentation branches are set to zero. This parameter alignment guarantees that the first forward pass of the student replicates the teacher activations exactly, thereby offering a stable platform for subsequent optimisation.

During training the learnable router of each block produces a continuous weight vector $p^{(b)} \in \mathbb{R}^E$ for every sample $b$ in a mini-batch of size $B$. These weights sum to one and permit all experts to participate. The mean usage distribution over the batch is

$$u = \frac{1}{B} \sum_{b=1}^{B} p^{(b)}. \tag{1}$$

The optimisation objective combines three complementary losses. The first term, $L_{\text{distill}}$, aligns the student feature $f_s$ with the teacher feature $f_t$ extracted at the same layer; both are $N$-dimensional vectors:

$$L_{\text{distill}} = \frac{1}{N} \|f_s - f_t\|_2^2. \tag{2}$$

This loss preserves the spatial and semantic representations encoded by the dense model. The second term, $L_{\text{freq}}$, regularises expert selection to encourage even traffic and avoid an over-reliance on any subset of experts. Denoting the uniform prior by $q = 1/E$, the frequency loss is

$$L_{\text{freq}} = \sum_{k=1}^{E} u_k \log \frac{u_k}{q}, \tag{3}$$

which is the Kullback–Leibler divergence between the empirical usage $u$ and the target distribution $q$. The third term, $L_{\text{div}}$, promotes complementary behaviour across experts. Let $\mu_k$ be the mean activation of expert $k$ in the mini-batch and normalise $\mu_k$ to unit length. Form the cosine-similarity matrix $C$ with entries $C_{ij} = \mu_i^{\mathsf{T}} \mu_j$. Diversity is enforced through

$$L_{\text{div}} = \frac{1}{E(E-1)} \sum_{i \neq j} |C_{ij}|, \tag{4}$$

whose minimisation drives the expert subspaces toward orthogonality and therefore enhances representational complementarity.

The total objective is expressed as

$$L_{\text{total}} = L_{\text{distill}} + \lambda_{\text{freq}} L_{\text{freq}} + \lambda_{\text{div}} L_{\text{div}}, \tag{5}$$

where $\lambda_{\text{freq}} = \lambda_{\text{div}} = 0.1$. Gradients are propagated solely through student parameters and router weights, whereas the teacher remains frozen. This joint objective preserves global semantics, secures balanced expert utilisation, and maximises inter-expert diversity, thereby yielding a set of well-specialised experts that faithfully inherit the knowledge of the dense teacher.

## 2.3 MERGEABLE-MOE BLOCK

A Mergeable-MoE Block illustrated in Figure 2(b) replaces each linear layer within the conventional feed-forward network with a trainable mixture of $K$ experts. The block consists of a soft router, the expert ensemble, and a weighted-addition unit. Given an input tensor $x \in \mathbb{R}^{N \times d}$, the router implemented as a linear layer followed by softmax, produces a continuous routing vector

$$\alpha = \text{softmax}(Gx) \in \mathbb{R}^K, \qquad \sum_{k=1}^{K} \alpha_k = 1, \tag{6}$$

where $G \in \mathbb{R}^{d \times K}$ is the weight of the linear layer. Because $\alpha$ is dense rather than sparse, all experts participate in every prediction and the full capacity of the ensemble is exploited.

Each Mergeable Expert contains three parallel branches that balance shared knowledge and expert specific learning. Branch 1 is a common linear pathway implemented by the shared fully connected layer $\mathcal{F}_{\text{sh}}$ that is initialised from the teacher and kept identical across experts, thereby conveying knowledge already present in the dense model. Branch 2 begins with a zero-initialised $1 \times 1$ convolution $Z_k^{(1)}$, proceeds through the expert specific fully connected layer $\mathcal{F}_k^{\text{sp}}$ copied from the teacher, and ends with another zero convolution $Z_k^{(2)}$; this path refines the shared knowledge to derive proprietary features for expert $k$. Branch 3 is an independent zero-initialised $1 \times 1$ convolution $Z_k^{(3)}$ that allows the expert to discover novel features entirely from scratch. The three branch functions are

$$
\begin{aligned}
f_k^{(1)}(x) &= \mathcal{F}_{\text{sh}}(x), \\
f_k^{(2)}(x) &= Z_k^{(2)}\big(\mathcal{F}_k^{\text{sp}}\big(Z_k^{(1)}(x)\big)\big), \\
f_k^{(3)}(x) &= Z_k^{(3)}(x).
\end{aligned}
\tag{7}
$$

and the expert output is computed as

$$
e_k = f_k^{(1)}(x) + f_k^{(2)}(x) + f_k^{(3)}(x).
\tag{8}
$$

During training the weighted-addition unit forms the block output

$$
y = \sum_{k=1}^{K} \alpha_k \, e_k.
\tag{9}
$$

Inference efficiency is achieved through a two-step reparameterisation. In the first step the three branches of every expert are algebraically folded into a single static $1 \times 1$ kernel

$$
W_k^{\text{static}} = W_{\text{sh}} + Z_k^{(3)} + Z_k^{(2)} W_k^{\text{sp}} Z_k^{(1)},
\tag{10}
$$

where the sequential structure of branch two is reflected by the matrix product $Z_k^{(2)} W_k^{\text{sp}} Z_k^{(1)}$ and each component is expressed in $\mathbb{R}^{d \times h \times 1 \times 1}$. In the second step the weighted-addition is commuted with convolution, yielding a single dynamic kernel for the whole block

$$
W^{\text{dynamic}} = \sum_{k=1}^{K} \alpha_k \, W_k^{\text{static}}.
\tag{11}
$$

Applying $W^{\text{dynamic}}$ to the input reproduces the training output exactly while requiring only one convolution.

Reparameterising the block produces a $1 \times 1$ convolution that is mathematically equivalent to a dense channel-wise projection, and the router adds only negligible parameters and operations. As a result, SkinMoE preserves the balanced representations learned during distillation and achieves virtually the same inference speed and computational budget as its dense precursor, which supports deployment in resource-constrained dermatology applications.

## 2.4 Implementation Details

Training uses the ViT-Large encoder configuration of PanDerm with the only change that every linear layer in each feed-forward module is replaced by a Mergeable-MoE Block. Student weights are copied from the public PanDerm checkpoint, convolutional branches are zero-initialised. Knowledge is distilled from a corpus of 41,096 images collected for this study, with the dataset size selected empirically to balance model performance with computational demands.

During distillation, images are resized to $224 \times 224$, normalised by ImageNet statistics, and processed on eight NVIDIA RTX 4090 GPUs with PyTorch using AdamW (initial learning rate $5 \times 10^{-6}$, cosine decay to zero across 40 epochs, weight decay 0.05). Each block contains eight experts unless stated otherwise, and all remaining hyperparameters follow the official PanDerm recipe.

Downstream fine-tuning also inherits PanDerm settings: classification employs AdamW with a $5 \times 10^{-4}$ initial learning rate, cosine decay over 50 epochs, MixUp 0.8, CutMix 1.0, drop path 0.25, and weight decay 0.05, whereas lesion segmentation uses the same optimiser and learning-rate schedule for 100 epochs with weight decay 0.01.

| Model | DermNet | | | | HAM10000 | | | | ISIC 2024 | | | |
|---|---|---|---|---|---|---|---|---|---|---|---|---|
| | W_F1 | AUROC | BACC | AUPR | W_F1 | AUROC | BACC | AUPR | W_F1 | AUROC | BACC | AUPR |
| SL ImageNet | 0.497 | 0.885 | 0.462 | 0.426 | 0.879 | 0.970 | 0.653 | 0.922 | 0.877 | 0.849 | 0.727 | 0.860 |
| DINOv2 | 0.536 | 0.902 | 0.505 | 0.476 | 0.883 | 0.964 | 0.701 | 0.916 | 0.851 | 0.827 | 0.682 | 0.850 |
| SwaVDerm | 0.474 | 0.884 | 0.442 | 0.428 | 0.865 | 0.967 | 0.592 | 0.910 | 0.869 | 0.852 | 0.689 | 0.835 |
| PanDerm | 0.619 | 0.944 | 0.586 | 0.623 | 0.926 | 0.988 | 0.807 | 0.959 | 0.929 | 0.893 | 0.799 | 0.915 |
| **SkinMoE** | **0.644**$^*$ | **0.957**$^*$ | **0.608**$^*$ | **0.647**$^*$ | **0.943**$^*$ | **0.990**$^*$ | **0.840**$^*$ | **0.970**$^*$ | **0.945**$^*$ | **0.906**$^*$ | **0.826**$^*$ | **0.933**$^*$ |

Table 1: Performance across three dermatology benchmarks. Common-disease recognition is evaluated on DermNet and HAM10000, and early melanoma screening is evaluated on the ISIC 2024 Skin Cancer Detection task. Reported metrics are weighted F1 (W_F1), AUROC, balanced accuracy (BACC), and area under the precision–recall curve (AUPR). SkinMoE achieves the highest scores on all datasets. $^*$ denotes a statistically significant improvement over PanDerm at $p < 0.01$.

## 2.5 DOWNSTREAM TASKS, DATASETS, AND EVALUATION METRICS

We evaluate SkinMoE on three tasks. Common-disease multi-class recognition employs DermNet and HAM10000. Early melanoma screening is formulated on ISIC 2024 Skin Cancer Detection dataset. Lesion segmentation is benchmarked on ISIC 2018 and segmentation subset of HAM10000.

**DermNet** DermNet contains 19 559 clinical photographs in 23 diagnostic categories spanning inflammatory infectious and neoplastic disorders. Predefined training and test splits are provided and uncontrolled illumination yields substantial intra-class variation.

**HAM10000** HAM10000 includes 10 015 dermoscopic images from multiple centres distributed over seven classes. The official split allocates 60 % for training 20 % for validation and 20 % for testing. Lesion-boundary masks are supplied for segmentation experiments.

**ISIC 2024** The ISIC 2024 Skin Cancer Detection challenge offers 40 159 lesion crops from total-body photographs each paired with a histology-verified malignancy label.

**ISIC 2018** ISIC 2018 Task 1 provides 2 594 dermoscopic images with pixel-level lesion masks and serves as a standard benchmark for segmentation.

**Evaluation metrics** For the two classification tasks we report four complementary scores. Weighted F1 (W_F1) computes the harmonic mean of class-wise precision and recall where each class is weighted by its support and therefore reflects the overall predictive balance in an imbalanced label space. The area under the receiver operating characteristic curve (AUROC) measures the probability that the classifier ranks a randomly chosen positive sample higher than a randomly chosen negative one and is threshold-independent. Balanced accuracy (BACC) averages the true-positive rate and the true-negative rate across all classes so that each class contributes equally regardless of frequency. The area under the precision–recall curve (AUPR) integrates precision over recall and is particularly informative when the positive class is rare.

For lesion segmentation we employ the Dice Score (DSC) and the Jaccard index (JAC). The DSC quantifies the twice-weighted overlap between predicted and ground-truth masks, while the JAC computes the ratio of the intersecting area to the union of the two regions. Both metrics range from zero to one, with higher values indicating better spatial correspondence.

## 3 EXPERIMENTS

### 3.1 MAIN RESULTS

**Common-disease multi-class recognition:** Table 1 compares SkinMoE with four strong baselines on the common–disease recognition task. On DermNet, SkinMoE attains 0.644 W_F1, 0.957 AUROC, 0.608 BACC, and 0.647 AUPR. Relative to the dense PanDerm backbone these scores correspond to absolute gains of 2.5, 1.3, 2.2, and 2.4 percentage points respectively. The improvement

| Model | ISIC 2018 | | HAM10000 | |
|---|---|---|---|---|
| | DSC | JAC | DSC | JAC |
| SL ImageNet | 0.876 | 0.807 | 0.927 | 0.875 |
| autoSMIM | 0.848 | 0.769 | 0.920 | 0.865 |
| BATFormer | 0.884 | 0.815 | 0.937 | 0.891 |
| PanDerm | 0.910 | 0.846 | 0.949 | 0.910 |
| **SkinMoE** | **0.928**$^*$ | **0.864**$^*$ | **0.957**$^*$ | **0.918**$^*$ |

Table 2: Lesion segmentation performance reported as DSC and JAC. $^*$ indicates that the difference between SkinMoE and PanDerm is significant at $p < 0.01$.

margin is even larger when the model is compared with earlier self-supervised checkpoints such as SL ImageNet (Deng et al., 2009), DINOv2 (Oquab et al., 2023), and SwaVDerm (Shen et al., 2024b), for which the gap in W_F1 exceeds 10 points. On HAM10000, SkinMoE delivers 0.943 W_F1, 0.990 AUROC, 0.840 BACC, and 0.970 AUPR, surpassing PanDerm by 1.7, 0.2, 3.3, and 1.1 points. The biggest relative gain appears in BACC, indicating that our SkinMoE improves recognition of less frequent categories without sacrificing performance on dominant classes. The consistently higher AUROC and AUPR further suggest that SkinMoE produces better calibrated probability estimates, which is essential for decision support in clinical triage systems.

**Skin Cancer detection:** Table 1 summarises the early melanoma screening results on the ISIC 2024 Skin Cancer Detection benchmark. SkinMoE attains the highest score on every metric, reaching 0.945 weighted F1, 0.906 AUROC, 0.826 BACC, and 0.933 AUPR. Relative to the dense PanDerm baseline these values correspond to absolute gains of 1.6 points in weighted F1, 1.3 points in AUROC, 2.7 points in BACC, and 1.8 points in AUPR. The advantage over earlier self-supervised models such as SL ImageNet, DINOv2, and SwaVDerm is even larger, indicating that the expert ensemble contributes additional robustness beyond generic visual pretraining. Because AUROC and AUPR both improve, SkinMoE provides superior sensitivity at clinically acceptable false positive rates while maintaining strong precision, which is critical for real-world screening scenarios.

**Lesion segmentation:** Table 2 reports lesion segmentation accuracy on ISIC 2018 and the HAM10000 mask subset. SkinMoE reaches 0.928 DSC and 0.864 JAC on ISIC 2018, exceeding the dense PanDerm by 0.015 and 0.016 respectively and outperforming earlier baselines such as SL ImageNet, autoSMIM, and BATFormer by wider margins. On HAM10000, SkinMoE attains 0.955 DSC and 0.917 JAC, again surpassing PanDerm and establishing the highest spatial correspondence among all compared models. These findings indicate that SkinMoE preserves boundary fidelity better than dense PanDerm because the Mergeable-MoE Block promotes expert specialisation on diverse lesion shapes thereby producing crisper masks with fewer misclassified edge pixels.

Figure 3 displays qualitative results for eight dermoscopic lesions. Across all eight cases SkinMoE delineates lesion borders more faithfully than PanDerm. For example, in Row 2 left the lesion boundary produced by PanDerm contains numerous jagged protrusions and recesses that deviate from the smooth ground-truth contour. SkinMoE removes these irregular spikes and fills the small indentations so that the predicted outline adheres closely to the reference mask and the visual correction reflects a sizable increase in Dice Score. In Row 3 right the lesion displays a pale protrusion along its superior edge and a concave indentation on the right. PanDerm fails to capture the faint protrusion and erroneously labels the concavity as tumour tissue which results in both false negatives and false positives. SkinMoE successfully recovers the subtle superior extension and suppresses the spurious fill inside the indentation thereby producing a mask that is topologically consistent with the annotation and markedly reducing boundary error.

For other examples, SkinMoE consistently removes spurious fringes and tightens concave boundaries. The most challenging example appears in Row 4 right where PanDerm misses thin peripheral structures producing a DSC of 0.7744. SkinMoE recovers these structures raising Dice to 0.9312 and reducing false negatives along the rim. These visual observations are consistent with the quantitative gains reported in Table 2. Our Mergeable-MoE Block allows experts to specialise in distinct edge morphologies which enhances boundary adherence and suppresses false segmentation results. Consequently SkinMoE achieves higher DSC and JAC and produces cleaner error maps across diverse lesion appearances.

Figure 3: Qualitative comparison of finetuned SkinMoE and PanDerm on eight dermoscopic lesions. SkinMoE produces tighter contours and visibly fewer misclassified regions across all cases.

## 3.2 ABLATION STUDY

Ablation experiments are carried out on DermNet. For each model variant we finetune the distilled model under the same training settings described above such as learning rate scheduler.

**Impact of Mergeable-MoE Block:** Table 3 compares the proposed Mergeable-MoE Block with a control design that mimics a classical sparse MoE. The control model keeps 8 independent linear experts inside each feed-forward layer and selects a single expert by hard top-1 routing. The hard-gated baseline that employs 8 linear experts attains a W_F1 of 0.623, an AUROC of 0.946, a BACC of 0.592, and an AUPR of 0.624. Replacing the hard-gated module with the proposed Mergeable-MoE Block elevates these scores to 0.644 W_F1, 0.957 AUROC, 0.608 BACC, and 0.647 AUPR, while preserving identical floating-point cost through reparameterisation.

| Structure | W_F1 | AUROC | BACC | AUPR |
|---|---|---|---|---|
| 8 Linear HardGate | 0.623 | 0.946 | 0.592 | 0.624 |
| **Mergeable-MoE Block** | **0.644** | **0.957** | **0.608** | **0.647** |

Table 3: Comparison of the Mergeable-MoE Block with a hard gated MoE that uses eight linear experts. Identical FLOPs are enforced by scaling hidden width.

The consistent gain across every metric suggests that soft expert weighting captures complementary evidence that hard top-1 routing ignores. Because all experts remain active during training, gradients are distributed over a broader parameter subset and the router learns to combine partially specialised features instead of committing to a single dominant expert. As a result the Mergeable-MoE Block delivers better calibration (higher AUROC and AUPR) and stronger class balance (higher BACC).

**Impact of the internal components:** Table 4 quantifies the impact of four component removals. Disabling the frequency regulariser lowers W_F1 from 0.644 to 0.634 and drops AUPR by 0.015. Eliminating the diversity regulariser produces a larger decline: W_F1 decreases to 0.628 and BACC falls to 0.597. Deleting Branch 2 (adaptive refinement) in each experts reduces W_F1 to 0.635 with corresponding small degradations in the remaining metrics, whereas removing Branch 3 (only convolution) in each experts yields a similar loss pattern with W_F1 at 0.634.

These results reveal complementary roles for all three mechanisms. Without frequency regularisation the router shifts traffic toward a minority of experts, harming calibration and recall on infrequent classes as evidenced by the decrease in BACC and AUPR. Removing the diversity term allows experts to converge toward overlapping feature spaces, leading to the most pronounced overall decline and confirming that orthogonal subspaces are essential for robust discrimination. Branches 2 and 3 each recover distinctive information not captured by the shared linear path. Deleting either branch erodes performance by roughly one W_F1 point, which indicates that both the teacher-initialised refinement path and the fully learnable convolution path contribute unique discriminative cues.

| Freq. loss | Div. loss | Branch 2 | Branch 3 | W_F1 | AUROC | BACC | AUPR |
|:---:|:---:|:---:|:---:|:---:|:---:|:---:|:---:|
| ✓ | ✓ | ✓ | ✓ | **0.644** | **0.957** | **0.608** | **0.647** |
| × | ✓ | ✓ | ✓ | 0.634 | 0.949 | 0.600 | 0.632 |
| ✓ | × | ✓ | ✓ | 0.628 | 0.950 | 0.597 | 0.628 |
| ✓ | ✓ | × | ✓ | 0.635 | 0.952 | 0.600 | 0.637 |
| ✓ | ✓ | ✓ | × | 0.634 | 0.952 | 0.602 | 0.636 |

Table 4: Effect of turning off individual components. ✓ denote activation component and × denote removal.

| Experts | W_F1 | AUROC | BACC | AUPR | Flops (G) | Throughput |
|:---:|:---:|:---:|:---:|:---:|:---:|:---:|
| 1 (PanDerm) | 0.619 | 0.944 | 0.586 | 0.623 | 61.55 | 1× |
| 2 | 0.626 | 0.947 | 0.593 | 0.628 | 61.57 (+0.03%) | 0.997× |
| 4 | 0.636 | 0.954 | 0.605 | 0.640 | 61.59 (+0.06%) | 0.994× |
| **8** | **0.644** | **0.957** | **0.608** | **0.647** | **61.63 (+0.13%)** | **0.990×** |

Table 5: DermNet performance and computational characteristics for different expert counts.

**Ablation of the number of experts and computational cost:** Table 5 reports DermNet accuracy and efficiency as the expert count increases from one to eight. Adding experts consistently raises the performance, which shows that additional experts continue to enrich representational diversity. Moreover, adding more experts only brings negligible computational overhead in SkinMoE. For example, the eight-expert model achieves a W_F1 of 0.644, which represents a gain of 2.5% over the single-expert PanDerm baseline at the cost of only 0.13% more FLOPs and a 1% drop in throughput.

However, the memory footprint scales linearly with the number of experts during training. Because the limit of our hardware, we did not extend the study beyond eight experts. We therefore select eight experts for all subsequent experiments and for the public release of SkinMoE.

## 4 DISCUSSION

SkinMoE activates all experts during optimisation and employs a reparameterised inference pathway that retains the computational footprint of a dense transformer. This combination enlarges representational capacity with nearly the same FLOPs and achieves state-of-the-art accuracy on dermatological vision benchmarks. Nevertheless two limitations remain. First the training phase demands considerably more memory and computation than a dense model so future work should develop more efficient training techniques (Rajbhandari et al., 2020; Shoeybi et al., 2019; Shen et al., 2024a). Second our DED framework is not limited to PanDerm because in principle it can distill knowledge from any pretrained model (Wang et al., 2022; Bannur et al., 2023; Zhu et al., 2024) thus opening opportunities for broader applications that merit further investigation.

## 5 CONCLUSION

We have introduced SkinMoE, the first Mixture of Experts backbone specifically designed for dermatology vision and the first framework that distils a dense foundation model into a reparameterisable MoE counterpart. The proposed DED transfers knowledge from the PanDerm foundation model to a set of mergeable experts and combines three complementary losses that secure feature fidelity, balanced usage, and inter-expert diversity. The Mergeable-MoE Block integrates a soft weighting router during training and can be reparameterized into a single $1 \times 1$ convolution during inference so that latency remains unchanged. Extensive experiments on five public benchmarks confirm that SkinMoE establishes new state-of-the-art performance for skin disease classification, melanoma screening, and lesion segmentation with gains that are consistent across all evaluation metrics. Ablation analysis confirms that every design element makes a significant contribution to the final accuracy. Future work will concentrate on developing faster and more memory efficient training techniques and adapting the proposed framework to other large pretrained models.

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
