# MoE Meets Reparameterization: Reparameterizable Mixture-of-Experts Model Enhances Dermatology Diagnosis via Dense-to-Experts Distillation

## 1 Related Work

**Mixture-of-Experts.** Early sparsely gated MoE layers demonstrated that conditional parameter utilisation can enlarge model capacity without proportional compute (Shazeer et al., 2017). GShard scaled this idea to $\sim 600$ B parameters in multilingual translation by integrating automatic tensor sharding with expert routing (Lepikhin et al., 2020). Switch Transformer further simplified routing to one-expert selection and trained a trillion-parameter language model with reduced communication overhead (Fedus et al., 2022). GLaM extended sparsity to multilingual web corpora and showed strong zero-shot generalisation (Du et al., 2022). Subsequent open-source efforts improved stability and transfer: ST-MoE introduced noise-based gating to equalise expert load (Zoph et al., 2022), OpenMoE provided modular implementations for continual pre-training (Xue et al., 2024), and MegaScale-MoE proposed pipeline parallelism for production-scale deployment (Jin et al., 2025). Auxiliary regularisers alleviate expert imbalance and collapse, for example expert-choice routing encourages uniform utilisation (Zhou et al., 2022), gating dropout reduces communication cost (Liu et al., 2022), and residual MoE blends dense and sparse paths (Wu et al., 2022a). Vision transformers later adopted sparse experts: V-MoE matched the accuracy of large dense ViT-G while saving FLOPs through token-level routing (Riquelme et al., 2021), $M^3$ViT co-designed accelerator-aware MoE blocks for multi-task learning (Fan et al., 2022), ViMoE explored depth-wise expert placement (Han et al., 2024), and Mobile V-MoE delivered mobile-scale latency by weight pruning and sparsity (Daxberger et al., 2023).

**Reparameterisation.** Structural reparameterisation trains over-parameterised multi-branch blocks then algebraically merges them into single-branch kernels for inference efficiency. ACNet first showed asymmetric convolutions can be fused into $3 \times 3$ filters for robustness (Ding et al., 2019). RepVGG stacked $3 \times 3$ convolutions during deployment and achieved $> 80\%$ ImageNet top-1 with VGG-style speed (Ding et al., 2021c). Diverse Branch Block generalised fusion to heterogeneous branches for improved generalisation (Ding et al., 2021b). ExpandNets applied linear over-parameterisation to compress student CNNs after training (Guo et al., 2020). ResRep decoupled "remembering" and "forgetting" filters to prune channels losslessly (Ding et al., 2021a). FastViT combined RepMixer token mixing with ViT blocks to reduce latency and improve accuracy (Vasu et al., 2023). RepFormer extended the idea to facial landmark transformers by fusing pyramid heads (Li et al., 2022).

**Dermatology Vision Models.** Esteva *et al.* trained an end-to-end CNN on $\sim 130,000$ clinical and dermoscopic images and reached dermatologist-level melanoma detection (Esteva et al., 2017). Subsequent reader studies confirmed AI superiority over clinicians in large cohort benchmarks (Brinker et al., 2019). Human–computer collaboration improved overall sensitivity by combining visual saliency with expert opinion (Tschandl et al., 2020). Reinforcement learning further aligned predictions with clinical risk preferences (Barata et al., 2023). Self-supervised pre-training on 75 M radiology and dermatology images advanced few-shot transfer (Azizi et al., 2021). Federated self-supervised contrastive learning preserved privacy across mobile dermatology assistants (Wu et al., 2022b). SwAVDerm mined unannotated community images via contrastive clustering and improved long-tail disease coverage (Shen et al., 2024). PanDerm combined slide-level and pixel-level supervision over $> 2$ M images plus 45 K pathology reports to set new state of the art across multiple benchmarks (Yan et al., 2025).