# OpenReview forum: "MoE Meets Reparameterization: Reparameterizable Mixture-of-Experts Model Enhances Dermatology Diagnosis via Dense-to-Experts Distillation"
_ICLR.cc/2026/Conference — ICLR 2026 Conference Withdrawn Submission_

### Official Review · Reviewer_uc9P · 2025-10-20

**Soundness:** 2
**Presentation:** 3
**Contribution:** 2
**Rating:** 4
**Confidence:** 5

**Summary:**

This paper proposes SkinMoE, a mixture-of-experts (MoE)-based foundation model for dermatology image analysis. SkinMoE includes (1) dense-to-experts distillation (DED), which distills knowledge from a pre-trained dense vision transformer to an MoE; (2) Mergeable-MoE blocks, which aims to improve inference efficiency through reparameterization; and (3) a soft weighting mechanism that activates all routing paths.

The proposed SkinMoE is evaluated on 4 dermatology datasets across 3 tasks (e.g., recognition and segmentation) and compared against prior foundation models.

**Strengths:**

* S1: This paper explores foundation models for dermatology image analysis, which is of interest to the community.

* S2: The integration of knowledge distillation, MoE, and reparameterization is well-motivated, aiming to balance model capacity and efficiency.

* S3: The proposed approach shows improved recognition and segmentation performance compared with baseline methods presented in the paper.

**Weaknesses:**

**Major Weakness**

* W1: The methodological novelty seems limited. Dense-to-sparse distillation, structural reparameterization, and soft weighting mechanism of MoEs have been studied in previous research [R1, R2, R3]. The key components of SkinMoE, incouding DED, mergeable-MoE, and soft weighting, seem to be straightforward adaptation of existing techniques to the dermatology domain.

* W2: This related work section lacks sufficient covery of  SOTA methods in dermatology image analysis and general foundation models. In addition, there are not sufficient discussion to differentiate SkinMoE from prior works.

* W3: The claim of model efficiency (e.g., Line 24) is not supported by quantitative evidence. Reporting computational metrics such as FLOPs, parameter count, or inference latency would be essential to substantiate this claim.

* W4: The experimental comparisons are limited. This paper does not include comparisons with existing models designed for skin lesion classification/segmentation. For example, the reported Dice scores of SkinMoE on ISIC 2018 (92.8) are not competitive with specialized methods, such as RA-Net (ISIC 2018 Dice: 93.25)[R4].

**Minor Weakness**

* MW1: Figure 1 is not referenced in the main text.

* MW2: Figure 2 is not very clear. What are those 'Feature' tensors come from? Do those colored boxes (yellow, purple, and orange) have special meanings?

[R1] Shu, Fangxun, et al. "LLaVA-MoD: Making LLaVA Tiny via MoE-Knowledge Distillation." ICLR 2024.

[R2] Vasu, Pavan Kumar Anasosalu, et al. "Fastvit: A fast hybrid vision transformer using structural reparameterization." ICCV 2023.

[R3] Mu, Siyuan, and Sen Lin. "A comprehensive survey of mixture-of-experts: Algorithms, theory, and applications." arXiv preprint arXiv:2503.07137 (2025).

[R4] Naveed, Asim, et al. "Ra-net: Region-aware attention network for skin lesion segmentation." Cognitive Computation (2024).

**Questions:**

**Primary  Questions/Suggestions**

* QS1: What are the differences between SkinMoE and prior works? What is the methodology novelty of the proposed framework?

* QS2: Could the authors provide efficiency analysis (e.g., GFLOPs, latency, parameter count) to support the claimed efficiency?

* QS3: Including comparisons with existing SOTAs for dermatology would make the experiment section stronger

**Minor Questions/Suggestions**

* MQS1: Please refine Figure 2 for better readability and explanation.

---

### Official Review · Reviewer_6tiW · 2025-10-26

**Soundness:** 2
**Presentation:** 2
**Contribution:** 2
**Rating:** 2
**Confidence:** 3

**Summary:**

The paper proposes SkinMoE, a dermatology-focused MoE foundation model built via “Dense-to-Experts Distillation” (DED) from a pretrained dense vision transformer (PanDerm). The key architectural component is a “Mergeable-MoE Block” whose experts and router are trained jointly with soft weighting (all experts participate), and which the authors claim can be reparameterized into a single 1×1 convolution at inference, making computation nearly invariant to the number of experts. Experiments on DermNet (and brief results on HAM10000 and ISIC 2024) report up to +2.5% Weighted F1 over PanDerm, with ablations indicating contributions from the reparameterizable block and the distillation losses.

**Strengths:**

- Clear problem framing: improving robustness of dermatology image analysis by replacing FFN linear layers with an MoE-style block and distilling from a domain foundation model (PanDerm).
- Practical efficiency intent: the block is designed to be reparameterized into a single 1×1 convolution at inference; soft gating ensures all experts participate, which can simplify training stability vs. sparse routing.
- Potentially useful engineering if rigorously validated and accompanied by solid efficiency profiling and fairness analyses.

**Weaknesses:**

Limited novelty of the core block
- The Mergeable-MoE Block is essentially multiple simple expert branches (shared FC path plus zero-initialized 1×1 conv branches) combined via a soft router and algebraically fused into a single 1×1 conv at inference. This closely mirrors established structural re-parameterization designs in CNN/ViT backbones (RepVGG, RepMLP, OREPA, FastViT) and prior soft/dense MoE routing (Soft MoE, DSelect-k, V-MoE), without a clear, isolated new principle or matched-baseline comparisons [1-7]. Actionable: Add head-to-head comparisons versus RepVGG/RepMLP/FastViT-style FFN replacements and established vision MoEs (V-MoE, Soft MoE, DSelect-k-based) under matched FLOPs/parameters, and ablate whether “MoE” is necessary compared to a single reparam’d branch.

“Foundation model” claim not substantiated by breadth or magnitude of benefits
- SkinMoE is not pretrained from scratch; it distills from PanDerm and uses a relatively small distillation corpus (41k images). Gains over PanDerm are modest (+2.5% Weighted F1 on DermNet), and hallmark FM properties (robust cross-domain transfer, zero-/few-shot capabilities, broad task coverage, fairness) are not demonstrated. This reads more like an architectural fine-tuning atop an existing FM rather than a new FM contribution [10][11][12]. Actionable: Either soften the FM claim or provide broader evidence (zero/few-shot; multi-task transfer beyond classification; strong OOD robustness; comprehensive fairness).

Incomplete baselines and comparisons
- Missing direct comparisons to strong vision MoE baselines (V‑MoE, Soft MoE, M3ViT, ViMoE) and to non‑MoE structural reparam FFN alternatives (RepVGG/RepMLP/FastViT). No top‑k router baseline despite explicitly positioning against it [7][5][9][13][1][2][4]. Actionable: Include these baselines under matched compute/params and report both accuracy and efficiency.

Narrow evaluation and fairness gap
- The paper has heavy reliance on DermNet, which is known to be imbalanced and skewed to lighter skin; no Fitzpatrick skin tone–stratified evaluation or bias analysis. External validation is limited (HAM10000 dermoscopic; ISIC crops) and missing calibration/clinical risk analyses. Clinical significance of small gains is not discussed [11][12]. Actionable: Add Fitzpatrick‑stratified metrics and subgroup error analyses; include multi‑center clinical datasets; report calibration (ECE/Brier) and decision‑curve analyses to contextualize clinical value.

[1] Ding, X., Zhang, X., Ma, N., Han, J., Ding, G., & Sun, J. (2021). RepVGG: Making VGG‑style convnets great again. In Proceedings of the IEEE/CVF Conference on Computer Vision and Pattern Recognition (pp. 13733–13742). https://openaccess.thecvf.com/content/CVPR2021/html/Ding_RepVGG_Making_VGG-Style_ConvNets_Great_Again_CVPR_2021_paper.html
[2] Ding, X., Zhang, X., Ma, N., Han, J., Ding, G., & Sun, J. (2022). RepMLPNet: Hierarchical Vision MLP with Re‑Parameterized Locality. In Proceedings of the IEEE/CVF Conference on Computer Vision and Pattern Recognition. http://openaccess.thecvf.com/content/CVPR2022/html/Ding_RepMLPNet_Hierarchical_Vision_MLP_With_Re-Parameterized_Locality_CVPR_2022_paper.html
[3] Hu, S., Wang, J., Zhang, Z., et al. (2022). Online Convolutional Re‑Parameterization. In Proceedings of the IEEE/CVF Conference on Computer Vision and Pattern Recognition. http://openaccess.thecvf.com/content/CVPR2022/html/Hu_Online_Convolutional_Re-Parameterization_CVPR_2022_paper.html
[4] Vasu, P. K. A., Gabriel, J., Zhu, J., Tuzel, O., & Ranjan, A. (2023). FastViT: A fast hybrid vision transformer using structural reparameterization. In Proceedings of the IEEE/CVF International Conference on Computer Vision. http://openaccess.thecvf.com/content/ICCV2023/html/Vasu_FastViT_A_Fast_Hybrid_Vision_Transformer_Using_Structural_Reparameterization_ICCV_2023_paper.html
[5] Gao, Y., Yu, X., Chen, T., et al. (2023). From sparse to soft mixtures of experts. arXiv:2308.00951. https://arxiv.org/abs/2308.00951
[6] Hazimeh, M., Abedalqader, M., Mazumder, R., et al. (2021). DSelect‑k: Differentiable selection in the mixture of experts with applications to multi‑task learning. In Advances in Neural Information Processing Systems.     https://proceedings.neurips.cc/paper_files/paper/2021/hash/f5ac21cd0ef1b88e9848571aeb53551a-Abstract.html
[7] Riquelme, C., Puigcerver, J., Mairal, J., et al. (2021). Scaling vision with sparse mixture of experts. In Advances in Neural Information Processing Systems. https://proceedings.neurips.cc/paper/2021/hash/48237d9f2dea8c74c2a72126cf63d933-Abstract.html
[8] Zhang, Z., Wang, Z., Liu, Y., et al. (2024). Dense training, sparse inference: Rethinking training of mixture‑of‑experts language models. arXiv:2404.05567. https://arxiv.org/abs/2404.05567
[9] M³ViT: Mixture‑of‑Experts Vision Transformer for Efficient Multi‑Task Learning with Model‑Accelerator Co‑Design. (2022). In Advances in Neural Information Processing Systems. https://proceedings.neurips.cc/paper_files/paper/2022/hash/b653f34d576d1790481e3797cb740214-Abstract-Conference.html
[10] PanDerm: A multimodal vision foundation model for clinical dermatology. (2025). Nature Medicine. https://www.nature.com/articles/s41591-025-03747-y
[11] Daneshjou, R., et al. (2024). Deep learning‑aided decision support for diagnosis of skin disease across skin tones. Nature Medicine, 30, 1052–1062. https://www.nature.com/articles/s41591-023-02728-3
[12] Skin type diversity in skin lesion datasets: A review. (2024). Current Dermatology Reports. https://link.springer.com/article/10.1007/s13671-024-00440-0
[13] ViMoE: An empirical study of designing vision mixture‑of‑experts. (2024). arXiv:2410.15732. https://arxiv.org/abs/2410.15732
[14] Cao, Y., Yang, S., et al. (2025). MoVE‑KD: Knowledge Distillation for VLMs with Mixture of Visual Encoders. In Proceedings of the IEEE/CVF Conference on Computer Vision and Pattern Recognition. https://openaccess.thecvf.com/content/CVPR2025/html/Cao_MoVE-KD_Knowledge_Distillation_for_VLMs_with_Mixture_of_Visual_Encoders_CVPR_2025_paper.html
[15] Zhang, H., et al. (2024). m2mKD: Module‑to‑Module Knowledge Distillation for Modular Transformers. arXiv:2402.16918. https://arxiv.org/abs/2402.16918
[16] Zhang, T., Dai, D., et al. (2021). EvoMoE: An evolutional mixture‑of‑experts training framework via dense‑to‑sparse gate. arXiv:2112.14397. https://arxiv.org/abs/2112.14397

**Questions:**

- Block specificity and alternatives: Can you precisely specify each expert branch’s layer shapes, activations, and normalization? Have you compared your block against non-MoE structural reparameterization baselines (e.g., RepVGG-/RepMLP-style replacements of FFN) under matched compute to test whether “MoE” is necessary?
- Router and specialization: Provide load-balancing metrics and expert specialization analyses. Does dense soft gating (all experts always active) reduce specialization compared to top-k? Add a direct top-k router baseline and DSelect-k/Soft MoE comparisons.
- Generalization and fairness: Evaluate across Fitzpatrick skin tones and diverse demographics; report per-tone metrics and error modes. Add multi-center clinical datasets beyond DermNet/HAM10000/ISIC and include calibration (ECE/Brier) and decision-curve analyses.
Claim scope: Given reliance on PanDerm and modest gains, reconsider the “foundation model” claim or broaden evidence to justify it (e.g., cross-task transfer, robustness under distribution shift, zero-/few-shot performance).

---

### Official Review · Reviewer_eiyW · 2025-11-02

**Soundness:** 2
**Presentation:** 3
**Contribution:** 3
**Rating:** 4
**Confidence:** 4

**Summary:**

The paper introduces **SkinMoE**, a Mixture-of-Experts (MoE) foundation model tailored for dermatology image analysis. It builds on a dense pretrained vision transformer (PanDerm) and transfers its knowledge into multiple experts using a Dense-to-Experts Distillation (DED) framework. Each linear layer in the transformer’s feed-forward network is replaced by a Mergeable-MoE Block, containing multiple soft-weighted experts that can be reparameterized into a single 1×1 convolution at inference time. This design allows all experts to contribute during training (unlike sparse top-k routing) while maintaining dense-model efficiency during inference.

Experiments on five public dermatology benchmarks **DermNet**, **HAM10000**, **ISIC 2018**, **ISIC 2024**, and the HAM10000 segmentation subset show consistent (but marginal) improvements over PanDerm across both **classification** and **segmentation** tasks. Some ablation studies have been made to support the author's architecture.

**Strengths:**

- The core idea of this work, distilling a dense ViT into a reparameterizable MoE, is interesting and has clear practical applicability in dermoscopic analysis.
- The authors include extended ablations, covering routing, loss functions, and influence of the number of experts considered, providing further support for the proposed approach.
- The authors also evaluate on five different datasets and two downstream tasks (segmentation and classification), indicating good robustness of the proposed method.
- Overall, the paper is well written, the main claims are sufficiently supported by the experimental evaluation, however some aspects of the evaluation need to be improved (as described below) to further strengthen the paper.

**Weaknesses:**

While the paper tests on several dermatology datasets, they all come from similar clinical or dermoscopic sources. Datasets like PAD-UFES-20 [R1] or PH2 [R2], include more diverse image modalities, including images acquired from  smartphones (PAD-UFES-20). Including these datasets would be important for further evaluating the robustness of the proposed method in real-world conditions. Since the paper emphasizes handling heterogeneity and different acquisition setups, omitting these datasets weakens the claims about generalization.

Additionally, the set of baselines considered is relatively narrow. Most are dense or self-supervised ViTs, but there is no comparison with other Mixture-of-Experts vision models (e.g., V-MoE [R3], or even smaller-scale MoE variants) that could further show whether the proposed Mergeable-MoE provides a significant advantage over existing MoE approaches. Additionally, vision encoders of lesion-metadata pre-trained models such as WhyLesionCLIP [R4] and SLIMP [R5] could also be considered as strong lesion classification baselines, while segmentation baselines such as SAM [R6] or Medical SAM [R7] would better contextualize the improvements in DSC/JAC. As it stands, the baselines make it difficult to judge how competitive SkinMoE really is. It is only clear that it improves performance compared to PanDerm.

Moreover, although the model shows consistent gains across datasets, the improvements are limited (1–3% over PanDerm). Considering the added complexity of the architecture and the significantly heavier training process, which raises doubts about scalability and practicality. The reparameterized inference is indeed efficient, but it only partially mitigates the high cost and difficulty of training such a model.

### Minor
There are a few small language and presentation errors that should be corrected. For example, in Figure 1, the caption uses “Comparation” instead of “Comparison.” A careful proofreading pass would help fix these minor mistakes and improve overall polish.

### References

[R1] Pacheco et al., "PAD-UFES-20: A skin lesion dataset composed of patient data and clinical images collected from smartphones". Data in brief, 32, 106221, 2020
[R2] Mendonça et al., "PH 2 - A dermoscopic image database for research and benchmarking". In international conference of the IEEE engineering in medicine and biology society (EMBC) (pp. 5437-5440), 2013
[R3] Riquelme et al., "Scaling vision with sparse mixture of experts". Advances in Neural Information Processing Systems, 34, 8583-8595, 2021
[R4] Yang et al. "A textbook remedy for domain shifts: Knowledge priors for medical image analysis." Advances in neural information processing systems 37 (2024): 90683-90713.
[R5] Christopoulos et al., "Skin Lesion Phenotyping via Nested Multi-modal Contrastive Learning". arXiv preprint arXiv:2505.23709, 2025
[R6] Kirillov et al., "Segment anything". In Proceedings of the IEEE/CVF international conference on computer vision (pp. 4015-4026), 2023
[R7] Ma et al., "Segment anything in medical images". Nature Communications, 15(1), 654 2024

**Questions:**

- How does the proposed method perform in datasets with diverse imaging modalities like PAD-UFES-20 and PH2?
- How does the proposed method perform in comparison to other MoE approaches?
- Have the authors tested the method on top of other architectures beyond PanDerm?

---

### Official Review · Reviewer_xhFM · 2025-11-03

**Soundness:** 2
**Presentation:** 2
**Contribution:** 2
**Rating:** 4
**Confidence:** 4

**Summary:**

This paper introduces SkinMoE, a Mixture-of-Experts backbone for dermatology image analysis. To leverage the rich features from a pretrained foundation model, the proposed method distills knowledge from a strong dense teacher (PanDerm) to finetune different sub expert modules with a soft weight fusion that encourages balanced and diverse experts. For model efficiency, the expert modules are reparameterized into a standard 1×1 convolution at inference.

**Strengths:**

1. The motivation is enough and good to solve the problem from PanDerm.
2. It shows better performance than PanDerm in different skin disease datasets.
3. The ablation study proves the effectiveness of the proposed modules.

**Weaknesses:**

1. The novelty is trivial. The proposed method only targets the problem from PanDerm while ignoring that most foundation models from other areas have already considered MoE strcuture and don't have such shortcomings. Besides, other modules mostly existed and were not new.
2. Though the motivation addresses heterogeneity and domain shifts, the experiments remain primarily within standard train/test splits. More explicit cross-dataset generalization (train on dataset A, test on B) or performance breakdowns across subgroups (e.g., skin tones) would strengthen the claim that SkinMoE better handles real-world heterogeneity.
3. The training efficiency should be discussed with more detailed quantitative results in the main paper. It seems the proposed structure is too complex.
4.  The exploration of the functions of each sub-expert should be investigated further. Currently, the whole method only utilizes MoE to solve the problem of PanDerm, but it doesn't show the deep reason why the expert module can achieve better.

**Questions:**

1. Add cross-dataset validation to prove robustness on domain shift, take [1] as the test set for evaluation on different skin disease types and skin colors.
2. Quantify training efficiency compared to PanDerm and discuss the training optimizations that are essential to make SkinMoE practical for real-world clinical usage.
3. The ablation study about loss hyperparameters should be included to analyze the model sensitivity.
4. Deep exploration of the reason why MoE achieves better performance.


[1] Wang, J., Hu, X., Zhang, Y., Almamy, D., Bamba, V., Koffi, K. A. S., ... & Yotsu, R. R. (2025). eSkinHealth: A Multimodal Dataset for Neglected Tropical Skin Diseases. arXiv preprint arXiv:2508.18608.

---

### Note · Authors · 2025-12-30

I have read and agree with the venue's withdrawal policy on behalf of myself and my co-authors.